

# Identification of tropical cyclones via deep convolutional neural network based on satellite cloud images

Biao Tong[1], Xiangfei Sun[2], Jiyang Fu[1,*], Yuncheng He[1,*] and Pakwai Chan[3]

[1]Research Center for Wind Engineering and Engineering Vibration, Guangzhou University, Guangzhou, China
[2]Guangdong Key Laboratory of Environmental Pollution and Health, School of Environment, Jinan University, Guangzhou, China
[3]Hong Kong Observatory, Hong Kong, China

*Correspondence to*: Jiyang Fu* (jiyangfu@ghzu.edu.cn); Yuncheng He* (yuncheng@gzhu.edu.cn);

**Abstract.** Tropical Cyclones (TCs) are one of the most destructive natural disasters. For the prevention and mitigation of
TC-induced disasters, real-time monitoring and prediction of TCs is essential. At present, satellite cloud images (SCIs) are utilized widely as a basic data source for such studies. Although great achievements have been made in this field, lack of concerns on the identification of TC fingerprint from SCIs have become a potential issue, since it is a prerequisite step for follow-up analyses. This paper presents a methodology which identifies TC fingerprint via Deep Convolutional Neural Network (DCNN) techniques based on SCIs of more than 200 TCs over the Northwest Pacific basin. Two DCNN models have been proposed and validated, which are able to identify the TCs from not only single-TC featured SCIs but also multi-TCs featured SCIs. Results show that both models can reach 96% of identification accuracy. As the TC intensity strengthens, the accuracy becomes better. To explore how these models work, heat maps are further extracted and analyzed. Results show that all the fingerprint features are focused on clouds during the testing process. For the majority of TC images, the cloud features in TC's main parts, i.e., eye, eyewall and primary rainbands, are most emphasized, reflecting a consistent pattern
with the subjective method.

## 1 Introduction

As one of the most destructive natural disasters, tropical cyclone (TC) can cause severe casualties and economic losses in TC-prone areas. The southeast coast of China is adjacent to the most TC-active ocean basin. Statistics show that an average of 30 TCs developing over the Northwest Pacific Ocean ever year, about one-third of which can make landfall in China,
resulting in an annual economic loss of $5.6 billion. With rapid development of urbanization in the coastal region of China, TC-induced disasters are expected to get even more severe.

To mitigate TC-induced disasters, real-time monitoring and forecasting of TCs activity are essential. To this end, various kinds of devices and techniques have been developed and utilized, such as radiosonding balloons, weather radar, wind profilers, airborne GPS-dropsonde, aircraft-based remote sensing equipment, and ever updating numerical models for
weather prediction. Since the 20th century, satellite started to be used in meteorology. Since then, satellite cloud images



(SCIs), which contain rich atmospheric information for investigating synoptic scale systems, have been more and more utilized as a basic data source for TC studies. An overwhelming advantage of SCIs against their counterparts is that they can be obtained effectively over each ocean basin in almost every desired period, making it possible to persistently and synchronously observe TCs from a global perspective.

With the assistance of SCIs, fruitful achievements have been gained. As one of the representatives, Dvorak technique is proposed for the systematic analysis of TCs, especially for identifying TC intensity (Dvorak, 1984). Dvorak technique is still under development in the following years (Velden et al., 1998; Velden et al., 2006; Olander et al., 2019). Up to date, most meteorological bureaus use the Dvorak technique to identify TC intensity. Another representative achievement is the Automated Tropical Cyclone Forecasting (ATCF) System, which is developed to predict track and intensity of TCs based on

a comprehensive usage of SCIs, detecting results from radiosonde, aircraft reconnaissance and other devices as well as numerical weather forecasting models (Miller et al., 1990; Sampson et al., 200; Goerss et al., 2000).

    For SCI-aided studies on TCs, it is prerequisite to identify TC fingerprint from the SCIs. Generally, the complex morphological characteristics of TCs and the coexistence of many interference factors, such as multiple cumulus clouds and continental background in the SCIs, make the task to be extremely challenging.

The most widely adopted method for TC identification from a SCI is the manual method. A series of fingerprint patterns are stratified in the Dvorak technique (Dvorak, 1984), which can be used empirically for TC identification. These patterns are summarized solely based on experiences about TC cloud shapes and their evolutionary characteristics. Apparently, application of the Dvorak technique relies greatly on users' experience and it also involves many manual manipulations, which makes it to be nonobjective and less efficient.

In this regard, increasing efforts have been made to identify TCs objectively, and several such methods have been developed, including the mathematical morphology method which exploits geometric transformations to extract morphological features on satellite images (Liao et al., 2011; Lopez-Ornelas et al., 2004; Han et al., 2009; Hayatbini et al., 2019; Wang et al., 2014), threshold method in which threshold filtering or segmentation processing is implemented (Di Vittorio et al., 2002; Wang et al., 2002; Sun et al., 2016; Lu et al., 2010), rotation coefficient method which applies vector

moments to represent the morphological characteristics of TC clouds (Geng et al., 2014). Despite the merit of objectivity, as these methods usually contain complex operations, they tend to be less user-friendly and more computationally expensive, especially for issues with a huge number of SCIs.

    In recent years, deep learning (LeCun et al., 2015; Schmidhuber, 2015; Zou et al., 2019) techniques, such as the Generative Adversarial Network (GAN), Recurrent Neural Network (RNN) and Convolutional Neural Network (CNN)

(Chen et al., 2020; Lee et al., 2020; Sun et al., 2021; Pang et al., 2021), have gained fast development. These techniques show overwhelming superiority against traditional approaches when dealing with data-intense prediction/recursive/classification problems. Thus, they provide a new way for studying TCs objectively and efficiently based on SCIs.



This paper presents a study on the identification of TC fingerprint via Deep Convolutional Neural Network (DCNN) techniques. As an abstract-feature-extraction oriented technique (Krizhevsky et al., 2012; Simonyan et al., 2014; Liu et al.,2019), DCNN is able to identify and classify various complex features involved in images in a highly generalized manner. Therefore, it turns to be more objective and convenient to identify TCs using this technique than using traditional methods. The reminder of this article is organized as follows. Section 2 states the methodology, where both the details of two specific DCNN models and the datasets are introduced. Section 3 presents typical results for the identification of TC fingerprint.

Main results and conclusions are summarized in Section 4.

## 2 Methodology statement

A flowchart of the methodology adopted in this study is shown in Figure 1. Two types of SCIs are available from open-source databases. For the first type which is most available, each SCI covers the entire Northwest Pacific Ocean basin (hereafter called NWPO image for short). Thus, it is not unusual that multi-TCs coexist in an NWPO image. By contrast, for

the other type, each SCI covers a much smaller region over the Northwest Pacific Ocean, in which there exists at most one TC (hereafter called L image). Although the NWPO images are more basic and more frequently utilized in practice, the L images are still employed in this study. The reason is twofold. First, it is necessary to preliminarily judge whether the DCNN technique is able to identify a specific TC effectively. For such feasibility testing, the L images are more appropriate. Second, it is much more difficult to identify TCs from an NWPO image than from an L image. Thus, the identification results

associated with the L images can provide useful reference to deal with the issues on the NWPO images.

In accordance to the features of the two types of SCIs, two DCNN models are proposed respectively. To improve the model performance, all SCIs are pre-processed initially (i.e., data augmentation and stratification of datasets). Then, the models are trained and validated via the SCI data. Visualization techniques (i.e., heat map) are further utilized to explore how the models work. Each link involved in the flowchart is detailed below.

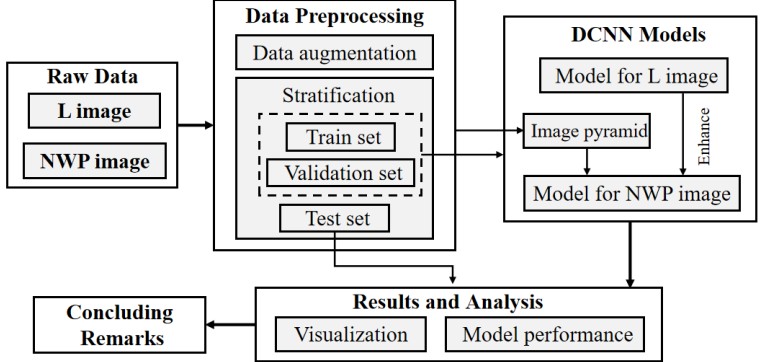


**Figure 1.** Flowchart of methodology



## 2.1 Data source

The L images refer to the high-resolution infrared SCIs that are captured by "Himawari-8", "MSSAT-1R" and other Japanese satellites over the Northwest Pacific Ocean. Each image contains 512×512 pixels in plane which cover a geographic area of about 20°×20°. These images corresponded to the snapshots at 1-hour intervals during the periods from 1-2 days ahead the formation of a TC to a couple of days after its dissipation in 2010-2019. Totally, 252 TCs were sampled during their whole life-cycle. Both the image data and corresponding label information, i.e., TC track and intensity, are available from the website of the National Institute of Informatics (NII) of Japan (http://agora.ex.nii.ac.jp/digital-TC/). Note that the intensity information of a TC is provided in a form of an integral multiple of 5 knots for this data source, and all intensity records are labeled as zero if they are below 35 knots. Owing to the poor quality of some SCIs and the absence of some TC label information, a limited number of the L images are discarded in this study, leaving about 47,000 valid images (the proportion between TC images and non-TC images is about 7:3) for the following analysis.

The NWPO images refer to the high-resolution infrared SCIs that were captured over the Northwest Pacific Ocean basin by geostationary satellites. Most of the images contain 1080×680 pixels in plane which cover a geographic area of 91°E-188°E and 3°S-55°N. These images corresponded to the snapshots at 3-hour intervals throughout the TC seasons in 2014-2019. Totally, about 160 TCs were sampled during their life-cycle. Each of the NWPO image has two formats: one with a colorful background of the earth's surface, and the other without it. The image data are available from the website of the Meteorological Satellite Research Cooperation Institute / University of Wisconsin-Madison (CIMSS) (http://tropic.ssec.wisc.edu/). As no corresponding label information is provided in the website, the one archived in the typhoon yearbooks as issued by the China Meteorological Administration (CMA) is adopted (http://tcdata.TC.org.cn/). In this study, ~15,000 valid images of this type are used, and the proportion between TC images and non-TC images is about 3:7.

## 2.2 Data preprocessing

### 2.2.1 Data augmentation

Although this study aims to identify a TC image from non-TC images, the identification performance of the proposed DCNN models potentially depends upon the TC intensity for TC images and the way how a TC is defined. Meteorologically, the intensity of a TC can be classified into several levels according to the maximum sustained wind speed at near ground in TC's inner region, i.e., tropical depression (22-33 knots), tropical storm (34-47 knots), severe tropical storm (48-63 knots), typhoon (64-80 knots), severe typhoon (81-99 knots) and super typhoon (≥100 knots). Because the intensity of a tropical depression is labeled as 0 knot for the L images in the data source, a SCI is regarded as a TC image throughout this study if it contains a TC storm whose intensity is labeled to reach or exceed the tropical storm level; otherwise, it is regarded as a non-TC condition.



To improve the model performance, it is expected that (i) there are sufficient samples for each of the typical categories (e.g., TC image or non-TC image) involved in the classification problem, and (ii) the numbers of samples are evenly

distributed among different categories. However, both the L image and NWPO image datasets suffer from an imbalanced distribution. Meanwhile, there are insufficient samples to cover each of the typical categories of TC images (e.g., with different TC intensity or TC numbers). For the L images, there are much fewer images for TCs with a higher intensity level, while for the NWPO images, there are much more non-TC images than those for TCs. To solve the above problems, two pre-processing techniques are adopted herein: down-sampling which is exploited when there are relatively more images involved

in a data type, and image transformation which is exploited for a data type with insufficient SCIs. As shown in Figure 2, five image-transformation modes are utilized, i.e., rotating 90°, 180° and 270° anticlockwise, as well as flipping horizontally and vertically, respectively. Through such transformation, the original image is able to generate 6 variations. On the one hand, down-sampling can be achieved by randomly selecting a certain portion of SCIs for a data type (i.e., TC images for L dataset and non-TC images for NWPO dataset). It is worth noting that using the image transformation technique is also beneficial

for improving the generalization ability of the DCNN models. However, owing to the rotating/flipping manipulations, some information involved in the image tend to be lost, e.g., rainbands spiraling anticlockwise in the Northern Hemisphere.

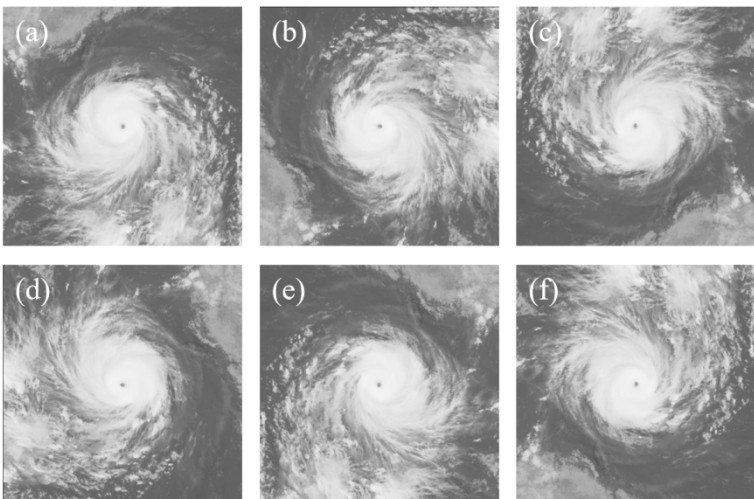

**Figure 2.** Image transformation: (a) original image; (b) rotating 90° anticlockwise; (c) rotating 180° anticlockwise; (d) rotating 270° anticlockwise; (e) horizontal flip; (f) vertical flip

**2.2.2 Stratification and standardization**

After data-augmentation processing, about 32,000 L images and 45,000 NWPO images are obtained. The datasets are divided into three sets: training set, validation set and test set. While the training set and validation set are respectively used to train and validate the DCNN models, the test set is used for overall model performance. In this study, both the L image and NWPO image datasets are stratified in such a way that the ratio of SCI numbers among training set, validation set and

test set is about 8:1:1.



All SCIs are then standardized in terms of pixel size and pixel value, to meet the requirements of the models for input information and to promote convergence during the training process. Each SCI is compressed to contain 100×100 pixels in plane for L images and 300×300 in plane for NWPO images. Meanwhile, all pixel values are normalized so that they are changed to be in a range of [-1, 1].

**2.3 DCNN model**

Convolutional Neural Network (CNN) is essentially a multi-layer perceptron. It can be used for classification and regression of images as well as automatic extraction of graphic features. In recent years, with the development of deep learning theory, DCNN have been further proposed based on the CNN techniques, which is usually regarded to possess a better performance than the CNN in terms of universality and accuracy.

Structurally, a DCNN model consists of several functional modules which can be combined in certain ways according to both internal logic and external requirements. Typical modules include convolutional layer, pooling layer, dropout layer, and dense layer. The convolutional layer contains a number of digital scanners, i.e., the convolutional kernels, whose sizes (called the kernel size) are fixed uniformly within the layer. This layer is used to read the input information of the model and obtain various potential features of the targets through convolution computation (filtering). Many convolutional layers may

be involved in a DCNN model. In principle, using more convolutional layers is beneficial for the model to generate more potential features of the targets. However, if there are too many convolutional layers, the model tends to suffer from gradient vanishing or explosion problems. Typically, a DCNN model consists of more convolutional layers than a CNN model. The pooling layer is mainly used to reduce the matrix information through a number of pooling operations, such as maximum pooling and average pooling. It is apparent that appropriate usage of the pooling layers can improve the computational

efficiency of the model effectively. The dropout layer (Srivastava et al., 2014) is used to maximize the efficiency of the neural nodes through eliminating unimportant features. Meanwhile, it also plays a role in avoiding over-fitting problems. Note that there are no dropout layers in a CNN model. At the end of the DCNN model is the dense layer (Jégou et al., 2017), which is used to flatten the information from previous layers and to estimate classification similarities through calculating a nonlinear function.

Figure 3 depicts the internal structure of the DCNN model for the NWPO images. The one for the L images is similar but somewhat simplified. Both models adopt a supervised learning strategy. The model for L images consists of 5 convolutional layers, 3 max-pooling layers, 2 dropout layers and 2 dense layers, while the one for NWPO images consists of 12 convolutional layers, 4 dropout layers, 2 max-pooling layers and 2 dense layers. For both models, the first convolutional layer serves as the input layer of the model, and the last dense layer serves as the output layer.





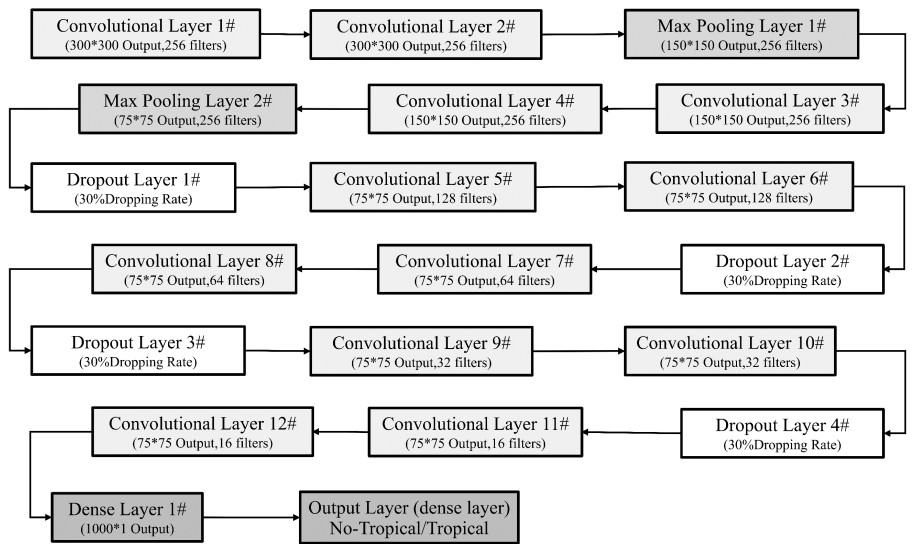

**Figure 3.** Detailed structure of the proposed DCNN model

Functionally, the input layer and the hidden layers cooperate to extract any potential features from SCIs for TC identification, while the output layer plays a role of judging and making decisions based on the extracted results. It is clear that to judging whether a SCI contains a TC essentially belongs to a binary classification problem. In this regard, the output

layer herein utilizes the binary cross-entropy loss function $L$ to quantify the inconsistence of the judgment (or training/prediction results) against the truth (i.e., data records):

$$L = -\frac{1}{N}\sum_{i=1}^{N}\sum_{c=1}^{M} y_i^c \cdot \ln[p(y_i^c)], \tag{1}$$

Where $y_i^c$ is the label of the $c$-th classification (1 for positive judgments and 0 for negative judgments) for the $i$-th SCI, N is the number of SCI samples, M is the number of categories, and $p(y_i^c)$ denotes the probability of the prediction associated

with $y_i^c$, which can be expressed via the softmax function:

$$p(y_i^c) = \frac{\exp(f_{y_i^c})}{\sum_{c=1}^{C}\exp(f_{y_i^c})}, \tag{2}$$

where $f_{y_i^c}$ is the original score of the model for prediction $y_i^c$, which is calculated by the output layer on the

basis of the output vector x (or the characteristic vector) from previous layers:

$$f_{y_i^c} = Wx + b, \tag{3}$$



in which $W$ represents the coefficient matrix which quantifies the weight for each element in $x$ during the

judging/prediction process, and $b$ is the bias vector.

Both $W$ and b should be determined through training. In this study, the stochastic gradient descent (SGD) method is

utilized to provide efficient estimation of the model parameters. Besides $W$ and $b$, there are also some hyper-parameters in

the DCNN model, including the number of neural network nodes and the learning rate. These parameters are usually preset

and adjusted empirically based on training results. Based previous tests, the model for L images in this study uses a learning

rate of 0.01, with a batch size of 64 and a training round number of 80. The settings of the model for NWPO images are

similar, but the batch size is changed to 8, and the number of training round becomes 100.

The model finally outputs a prediction value for each SCI, which is in the range of 0-100%. If the value exceeds 50%,

it is judged that the SCI belongs to a TC image; otherwise, the SCI is classified as a non-TC image.

**2.4 Model performance**

The parameters of accuracy, precision rate, recall rate and F-measure are conventionally adopted to indicate the performance

of a DCNN model. In this study, a SCI is classified as a positive sample if it contains a TC; otherwise, it is marked as

negative. Accordingly, "accuracy" is defined to represent the percentage of correctly classified (both positive and negative)

samples in the dataset; "precision rate" indicates the proportion of correctly identified positive samples in all positive

predictions, while "recall rate" represents the percentage of correctly identified positive samples in all positive samples. The

latter two parameters differ with each other in that precision rate highlights the performance in terms of not making

misjudgments, while recall rate focuses on the ability of avoiding the omission of positive predictions. During the training

and validating processes, the training accuracy and validation accuracy are usually compared in real time to determine

whether an over-fitting problem occurs.

$$Accuracy = \frac{N_{TP} + N_{TN}}{N_{TP} + N_{TN} + N_{FP} + N_{FN}} \qquad (4)$$

$$Precision = \frac{N_{TP}}{N_{TP} + N_{FP}} \qquad (5)$$

$$Recall = \frac{N_{TP}}{N_{TP} + N_{FN}} \qquad (6)$$

where $N_{TP}$ represents the number of true positive predictions, $N_{TN}$ is the number of true negative predictions, $N_{FP}$ is the

number of false positive predictions, and $N_{FN}$ is the number of false negative predictions.

F1-score is another indicator for the performance of models associated with classification problems. For binary

classification problems, F1 can be expressed as the harmonic mean of precision and recall:





$$F1 = 2\frac{Recall \times Precision}{Recall + Precision} = \frac{2N_{TP}}{2N_{TP} + N_{FN} + N_{FP}} \tag{7}$$

In reference to classification problems, it is not unusual that the numbers of samples associated with different categories vary significantly. Under such conditions, it becomes inappropriate to evaluate the model performance via a single value of the above parameters. For binary classification problems, the so-called Receiver-Operating-Characteristic (ROC) curve and Precision-Recall-Curve (PRC) are often adopted to provide more intuitive evaluation results (Powers et al., 2020; Hanley et al., 1982; Molodianovitch et al., 2006; Saito and Rehmsmeier, 2015).

A ROC curve compares the True-Positive-Rate (TPR, $=N_{TP}/(N_{TP}+N_{FN})$) against the False-Positive-Rate (FPR, $=N_{FP}/(N_{FP}+N_{TN})$), usually with FPR as the abscissa and TPR as the vertical coordinate. As the classification results depend upon how the prediction criterion (i.e., probability threshold of positive predictions) is defined, one can obtain a series of TPR and FPR values by selecting different threshold levels during the prediction process. In general, the smoother the ROC curve is, the better the classifier becomes. One can further use the so-called Area-Under-Curve (AUC), which expresses the area demarcated by the ROC curve in the coordinate system, to quantify the accuracy of prediction results.

PRC is similar to the ROC curve in form, but it compares recall rate (as the abscissa) and precision (vertical coordinate). A preferred classifier should correspond to a smooth PRC located toward the top right corner of the coordinate system. Usually, PRC works better than ROC for the cases with severe imbalance of samples between the positive and negative categories.

On the other hand, to improve the robustness of the model performance, the cross-validation strategy (Kohavi, 1995) is often exploited. As introduced previously, the original data in this study are stratified into 10 parts, with 9 parts used as training/validation set and 1 part as test set. By using the cross-validation strategy, the data can be trained and tested upmost for 10 times.

**2.5 Model visualization**

Basically, a DCNN model can be regarded as a "black box", since it is very difficult to explore the working mechanism of the model in a way that can be understood by human beings. In recent years, great efforts have been made to better understand how a DCNN model works internally.

As introduced previously, convolutional layers are able to extract various features from SCIs, which actually can be visualized by the so-called feature maps. As an example, Figure 4 shows the visualization results of the outputs from all convolution kernels involved in the first 4 convolutional layers (i.e., Conv1, Conv2, Conv3, Conv4) of a DCNN model. There are 256 convolution kernels in the Conv1 layer. Accordingly, they correspond to 256 feature maps. It is seen that some of the feature maps (e.g., feature map 3) are very similar to the input image except that many marginal details in the input image are filtered in the feature map. For deeper convolutional layers, the feature maps become more abstract (e.g., feature map 34), and some convolution kernels may fail to generate valid feature maps (e.g., the black squares in Conv2 and Conv3).

To clarify how the extracted features from convolutional layers influence the prediction results of a DCNN model, the Class-Activation-Map (CAM) technique is proposed (Zeiler et al., 2014; Selvaraju et al., 2017; Chattopadhay et al., 2018),

245  which essentially aims to generate a "heat map" through computing the weighted sum of all activation maps associated with the convolution kernels in a convolutional layer. Since a heat map expresses the extracted fingerprint features in a colorful form in the original image, one can then distinguish the emphasized features by the model from others. Then, the CAM technique has been developed into more advanced versions, e.g., Grad CAM and Grad CAM++. Because the Grad CAM++ technique is able to focus on fingerprint patches more accurately and meanwhile to cover multiple targets, this study adopts

250  this technique in the following analysis.

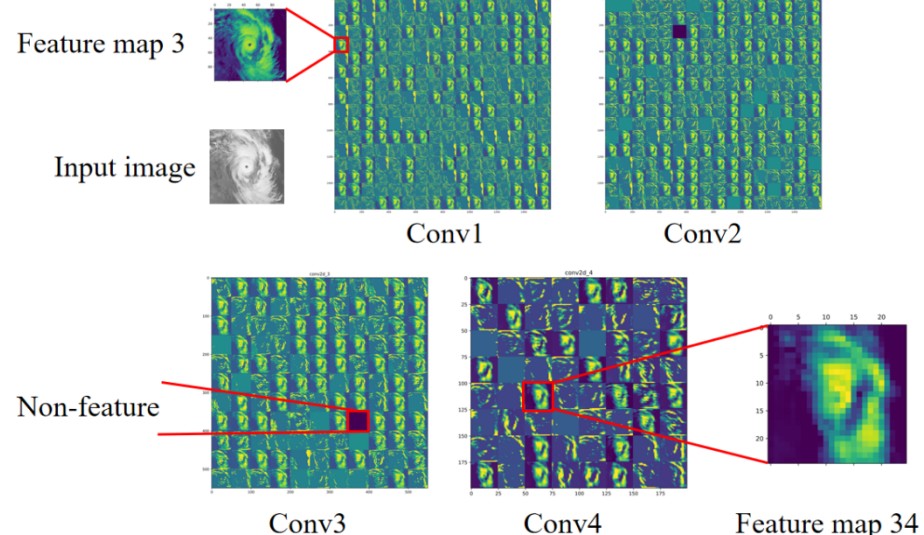

**Figure 4.** Visualization of outputs from the convolution kernels involved in the first 4 convolutional layers of a DCNN model

## 2.6 Computational platform

255  The DCNN models and supervised learning algorithms were coded using Python 3.7 in conjunction with the Keras 2.2.4 and Tensorflow 1.11.0/2.0.0 packages. The training process was executed by a combine usage of NVIDIA GeForce RTX 2080Ti×4 GPU, parallel computing management software CUDA (v10.0) and acceleration library cuDNN (v7.3.1.20, v7.6.0.64).





## 3. Results

### 3.1 Results for L images

#### 3.1.1 Overall performance

The 10-fold cross-validation strategy is employed to examine the robustness of the model performance. Figure 5 depicts the 10 evolutional curves of prediction accuracy during both training (TG) and validation (VG) processes for L images. As demonstrated, the training accuracy increased rapidly within the first 40 epochs (from 61% to 97%) and then leveled off at a considerably high level (~100%), which demonstrates the good convergence of the model during the training process. Results for the validation process were similar. Most of the 10 accuracy curves varied insignificantly and got stabilized at a high accuracy (>90% after 40 epochs). These results reflect the robustness of the model performance among different groups of samples. The high accuracy during both training and validation processes also reveals that the proposed model does not suffer from over-fitting problems. This should be partially attributed to the usage of dropout layers in DCNN models. Noted that using dropout layers tends to slow down the convergence rate of the model slightly at the beginning of the training process, as reflected by the results within the first several epochs in Figure 5 (dropout layers do not participate in work during the validation process). However, it does not influence the overall convergence rate (therefore the training efficiency) of the model noticeably.

Results from the training and validation processes show that the prediction performance of the model driven by dataset TG-1 (hereafter referred to as TG-1 model) agreed best with the average performance obtained via the 10-fold cross-validation strategy. As a result, this specific model is expected to be able to generate more representative predictions than others. Therefore, it was adopted for the analysis during the testing process.

Table 1 summarizes the prediction performance of the TG-1 model during the testing process. The values of prediction accuracy, precision, recall ratio and F1 are 96.43%, 96.72%, 94.49% and 95.59%, respectively. Figure 6 depicts the associated ROC and PRC curves. Based on the ROC curve, the AUC is calculated to be 99.10%. The high values of these parameters, especially the one of AUC, and the favorable features of ROC (smooth) and PRC (located toward the upper right) curves suggest the overall good performance of the proposed model for L images during the testing process.

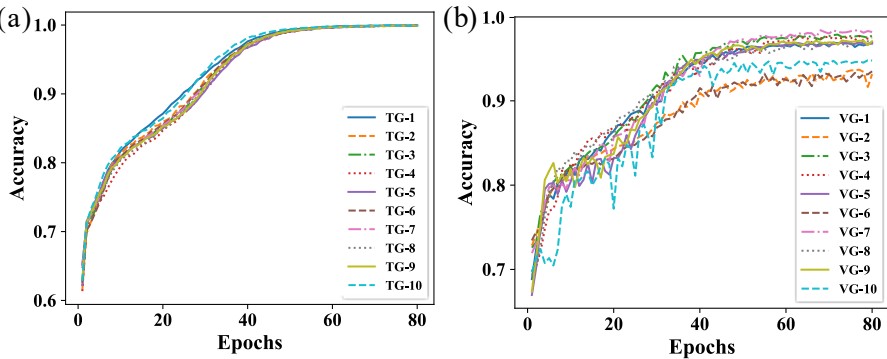





**Figure 5.** Evolutional curves of the prediction accuracy of the proposed DCNN model for L images: (a) training process; (b) validation process

**Table 1.** Prediction performance of TG-1 model during testing process

| Parameter | Accuracy | Precision | Recall ratio | F1-Score | AUC |
|---|---|---|---|---|---|
| Value | 96.43% | 96.72% | 94.49% | 95.59% | 99.10% |

**Figure 6.** ROC and PRC curves of TG-1 model for L images during testing process

To examine the potential influence of TC intensity on the prediction performance of the proposed model, Figure 7 exhibits the predicted probability for each of the TC images with varied intensity levels in the testing dataset. As demonstrated, although there were a limited number of misjudged samples for the cases with low intensity levels, very few (only 2) samples were misjudged for the cases with higher intensity levels (i.e., >70 knots), which reflects the proposed model has a much better ability to identify stronger TCs from SCIs. This is understandable, since more intense TCs tend to possess more typical fingerprint features that can be better detected and recognized by the classifier.

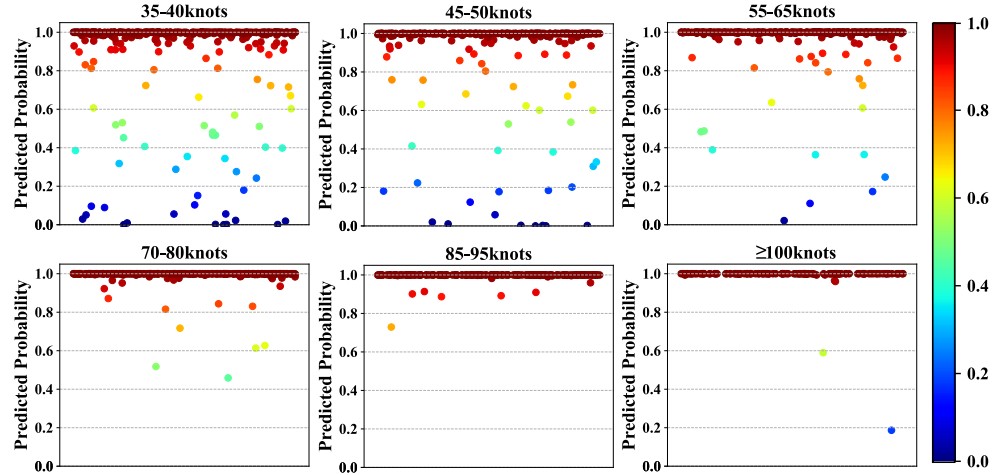

**Figure 7.** Predicted probability for each of TC images with varied intensity levels in the testing L image dataset via TG-1 model



Figure 8 depicts some typical SCIs of TCs with varied intensity levels that were misjudged as non-TC images via the TG-1 model during the testing process. Main TC structures are demarcated by red dash squares in the figure. By contrast,

Figure 9 depicts some non-TC images but misjudged as TC images. Noted that some information (e.g., rainbands spiraling anticlockwise in the Northern Hemisphere) for these images has been lost due to the rotating/flipping manipulations involved in the data pre-processing stage.

It is a bit strange that the SCI with a super typhoon (Figure 8(f)) was misjudged as a non-TC image. Scrutinizing TC images shows that the number of SCIs for super typhoons is very limited, and almost all the super typhoons contain a calm

eye at the center of the storm. However, there is no such a distinct TC eye in Figure 8(f). Instead, the image looks much similar to some of the non-TC cases, e.g., Figure 9(a, c). It is most likely that the model failed to identify this case due to the lack of appropriate training samples. Figure 8(c) corresponds to another kind of misjudged TC images. Although this TC was labeled as a severe tropical storm (55 knots), it stayed at a rapidly decaying stage around and after landfall. The other cases in Figure 8 may be regarded as the third kind. For these TCs, the morphological structures of TC cloud become too

multifarious (thus, the training samples turn to be insufficient) and irregular to be distinguished accurately by the model from those for non-TC cases as shown in Figure 9.

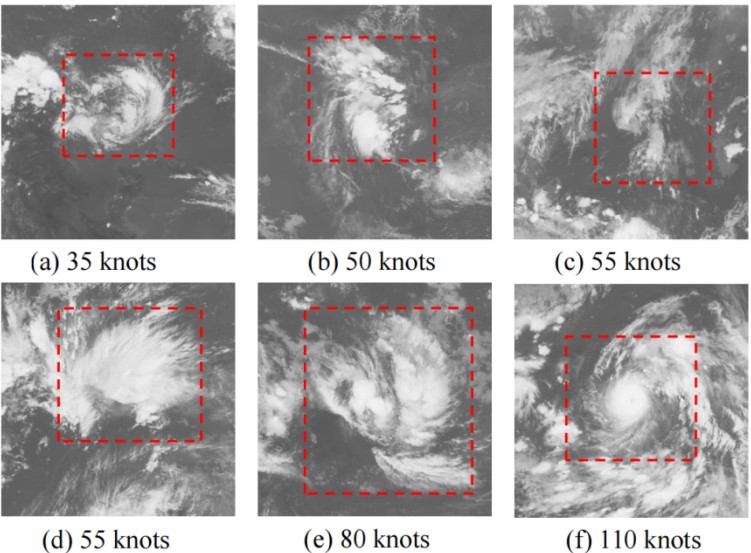

**Figure 8.** L images of TCs with varied intensity misjudged as non-TC images during testing process (main TC structures are demarcated by red dash squares)


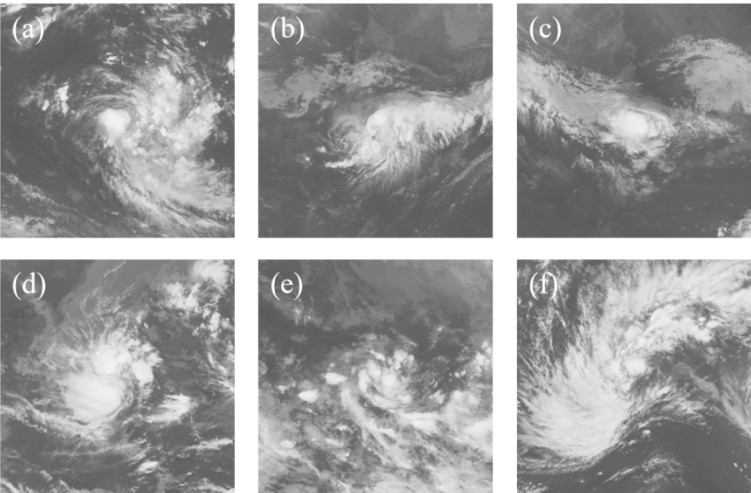

**Figure 9.** L images of non-TCs misjudged as TC images during testing process


In reference to the results shown in Figure 9, the failure of the model may be attributed to two main reasons. First, as tropical depression storms are not regarded as TCs in this study (refer to Section 2.2.1), some misjudged non-TC images actually contain a tropical depression. As one can imagine, there should be no evident differences between tropical

depressions and TCs with a slightly higher intensity level (e.g., tropical storm). Second, for some "true" non-TC images, the morphological characteristics of clouds are so similar to those for TCs that it is difficult to distinguish them effectively. Whatever the reason, more training samples that cover each type of the misjudged SCIs is needed so that the model can be trained to perform more accurately.

### 3.1.2 Model visualization

Figure 10 depict the heat maps (color pictures) of the TG-1 model for some successfully identified TC images. Corresponding SCIs (grey pictures) are also depicted for reference. In principle, during the identification process, the DCNN model would pay more attention to the graphic features that correspond to the areas depicted in a warmer color in the heat map. Thus, one can find out what the model most concerns with in a SCI intuitively. Results in the figure suggest that all the highlighted features are focused on clouds especially on large masses of clouds, which is consistent with the way adopted by

mankind. However, compared to the conditions for other identification issues (e.g., with a cat or dog) where usually only a few detail features (e.g., mouth, nose/whiskers or ears) within a small portion of the picture are emphasized, there were much more highlighted features that were scattered throughout the image in this study. The above difference reflects the complexity of identifying TCs from SCIs.

Basically, the heat maps can be categorized into two types, i.e., Type Ⅰ and Type Ⅱ as shown in Figure 10. For Type

Ⅰ which accounts for over 80 percent of the total results, the main body of TC cloud (corresponding to TC eye, eyewall and primary rainbands) was identified as the most typical features. Within the TC body, the inner portions (i.e., eye and eyewall)



received even more concerns than the outer region. These findings are consistent with traditional knowledge about the inner structures of a TC and their storm-relative distributions. By contrast, results in Type Ⅱ demonstrate a different pattern. The main body of TC cloud was no longer highlighted most significantly, although it was still regarded as one of the main

fingerprint features. For this type, it is still unclear whether the model is able to identify the TC successfully with an unknown but correct method, or it only happened to make a right prediction but in a wrong way. Thus, more advanced visualization techniques are required to further explore how a DCNN model works internally.

| Type Ⅰ | Type Ⅱ |
|---|---|
| Tropical storm | Tropical storm |
| Severe tropical storm | Severe tropical storm |
| Typhoon | Typhoon |
| Severe typhoon | Severe typhoon |





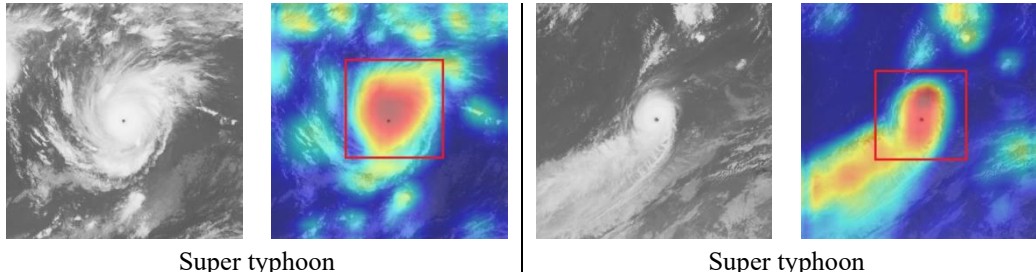

| Super typhoon | Super typhoon |

**Figure 10.** Heat maps (color pictures, TC is demarcated by red rectangular; hereafter) of some successfully identified TC images compared with associated L images (grey pictures; hereafter)

Figure 11 examines the heat maps for some non-TC images but misjudged as TC images (i.e., false positive predictions). Basically, these results show a high degree of similarity to those for TCs with a weak intensity level in Figure 10. In fact, a large number of such false positive predictions correspond to the dissipative process of a TC during or after landfall when the intensity of the storm was decreased to a level below 35 knots and the formerly huge TC cloud was fragmented. It is possible that the DCNN model is able to correlate the patches of clouds with a decaying TC.

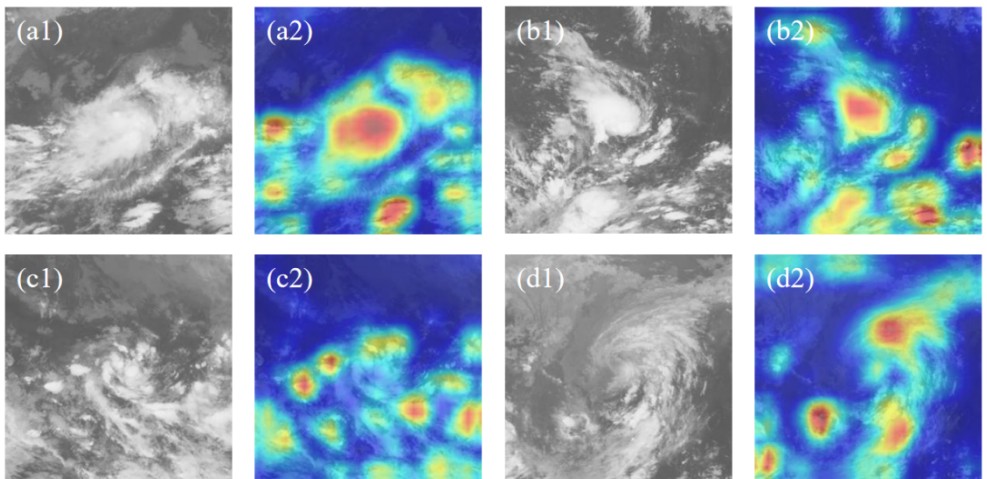

**Figure 11.** Heat maps of some non-TC images but misjudged as TC images compared with associated L images

### 3.2 Results for NWP images

### 3.2.1 Method improvement via image pyramid technique

The NWPO images can be analyzed in a similar way to the one for L images. However, results from previous attempts show that the fingerprint features highlighted in the heat maps of the DCNN model (Figure 3) could not be focused on TC clouds, although the prediction accuracy of the model was pretty high. This should be attributed to the fact that the TC structure is too small compared to many other graphic features involved in an NWPO image (e.g., coastal lines and continent/peninsulas) and the model is able to correlate the label of the image with such unexpected features rather than the TC clouds.



To impel the model to focus on cloud features during the identification process, the model was trained by the NWPO
images in conjunction with the L images. However, the obtained results of heat maps were still abnormal. What is worse, the
problem of decreased prediction accuracy occurs. Attempts were also made to train the model by NWPO images in
conjunction with the zoom-in views of TCs that were extracted from the NWPO images, but the results were similar to those
in the former case. The unacceptable performance of the model may be explained as follows. As mentioned previously, the
NWPO image covers a much larger region (97°×52°) than the L image (20°×20°). As a result, the scale of TC structures in
NWPO images turns to be considerably smaller than that in L images, which makes it difficult for the model to correlate the
TC features in NWPO images with those in L images effectively.

Finally, the image pyramid (IP) technique was adopted and the results turned to be satisfactory. The IP technique was
firstly proposed to solve classification problems for targets with varied scales in the classifier, such as face detection and
recognition (Zhang et al., 2016). With the development of feature pyramid networks (FPN), it was then used to deal with
problems with small targets (Lin et al., 2017). Due to the great convenience in operation and high efficiency in performance,
this technique has been increasingly exploited in the field of computer vision.

Table 2 and Figure 12 summarize the changes before and after using IP technology. It can be clearly seen from Table 2
that all parameters are only slightly improved after IP technology is used. However, it can be obvious from Figure 12 (b) and
(c) that the weight range of the model after using IP technology basically covers TC clouds or TC-like clouds, while heat
maps without IP technology show relatively strange attention.

**Table 2.** Comparison of indicators before and after using IP technology

| Parameter | Accuracy | Precision | Recall | F1-Score |
|-----------|----------|-----------|--------|----------|
| Without IP | 95.30% | 93.88% | 96.57% | 95.21% |
| With IP | 96.62% | 94.73% | 98.65% | 96.65% |

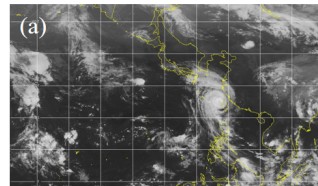
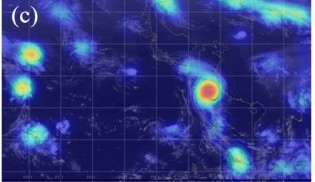

Raw image      Before IP      After IP

**Figure 12.** Heat map before and after using IP

Image pyramid is actually a cascade structure for multi-scale representations of an image. It consists of a series of
pictures that are derived from the same original image but with varied characteristic scales for highlighted targets. As an
example, Figure 13 illustrates the realization process for the image pyramid of a TC image. The original image (herein called
large picture) consists of 1080×1080×3 pixels. One can extract a portion (containing 512×512×3 pixels, 420×420×3 pixels,
etc.) of the image to generate a zoom-in view of the TC (herein called medium picture). In this study, a random extraction
stratify was adopted. Similarly, a second stage zoom-in view (containing 256×256×3 pixels, 200×200×3 pixels, etc.) can be
derived from the original picture (herein called small picture). The large, medium and small pictures are then normalized in



terms of size (i.e., to have same number of pixels). A combination of the normalized several pictures forms a multi-level image pyramid of the original picture. As can be seen, the TC in these pictures is expressed in varied scales and resolutions.

Using the IP technique, the input samples of NWPO images for the DCNN model turned to be ~135,000, and the proportion among the numbers of small, medium and large images approached to 1:1:1. The model was then trained and
validated based on the samples following the way as introduced previously. Note that only the large images (~4600 samples) were tested and analyzed during the testing process.

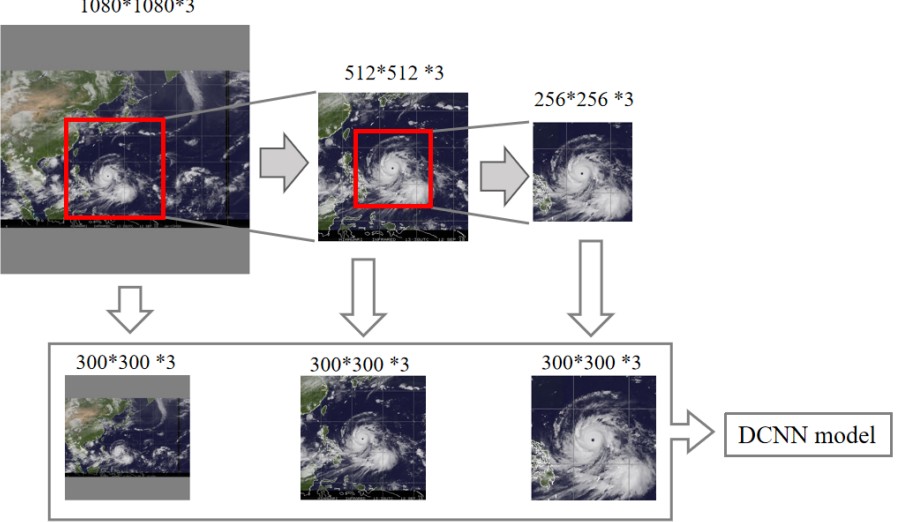

**Figure 13.** Flowchart for the realization of image pyramid of a TC image

**3.2.2 Overall performance**

Because there are many more NWPO images than L images, the prediction performance of the proposed model for NWPO images tend to be more robust. Thus, when using the 10-fold cross-validation strategy to train and validate the model, only three times were conducted. Figure 14 shows the evolutional curves of the prediction accuracy during both the training and validation processes. Following the analytical method adopted for related results of L images, it is found that the performance of the model for NWPO images is satisfactory in terms of convergence, robustness and anti-overfitting. The
training accuracy within 100 epochs reached 99.99%, compared to 95.37% for the validation accuracy. These results demonstrate the effectiveness of using the IP technique to deal with identification problems with small targets. Although the performances for the three training/validation processes were similar, the one associated with TG-1 dataset was comparatively more consistent with the ensemble-mean performance. Thus, the model trained by this dataset, i.e., TG-1 model, was adopted during the testing process.





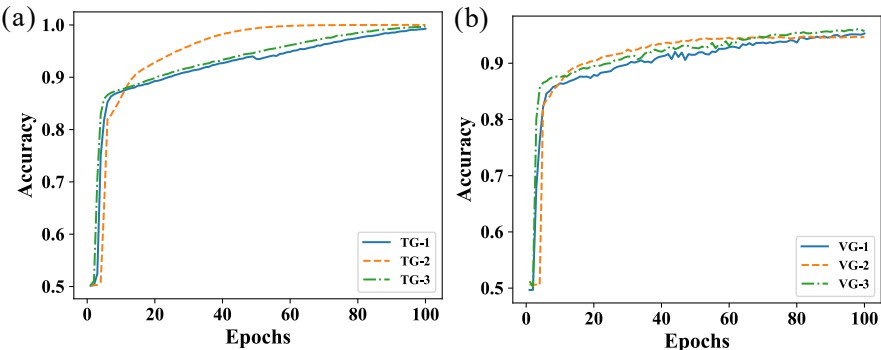

**Figure 14.** Evolutional curves of the prediction accuracy of the proposed DCNN model for NWPO images: (a) training process; (b) validation process

The prediction performance of the TG-1 model during the testing process is detailed in Table 3 and Figure 15. The results for all studied indexes except for precision and AUC are found to be better than their counterparts for L images (refer to Table 1 and Figure 6), although there exist many unfavorable factors in NWPO images for the performance of the model. The reason is twofold. First, using the IP technique makes it effective to train the model with NWPO images. Second, training the DCNN model with more samples helps to promote the prediction accuracy.

Table 3. Prediction performance of the TG-1 model during testing process

| Parameter | Accuracy | Precision | Recall ratio | F1-Score | AUC |
|---|---|---|---|---|---|
| Value | 96.62% | 94.73% | 98.65% | 96.65% | 98.90% |

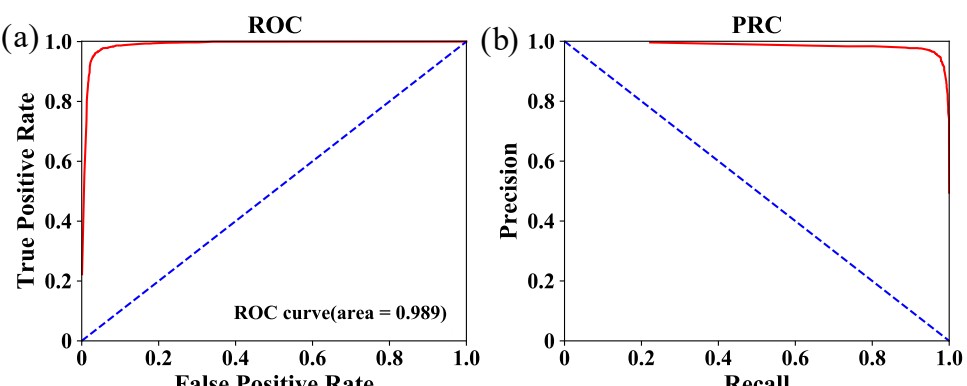

**Figure 15.** ROC and PRC curves of the TG-1 model for NWPO images during testing process

Given the better performance of the model for NWP images in terms of accuracy and recall ratio than that for L images, the relatively lower precision herein reveals that there were relatively more false positive predictions than those for L images. To better understand this issue, Figure 16 depicts some selected non-TC images but misjudged as TC images and associated heat maps. It is found that the majority of such false positive predictions corresponded to images whose cloud characteristics were similar to those of SCIs with a TC at a low intensity level (refer to Figures 8-9). As discussed previously, there are two





possibilities. First, the image indeed involves a TC, but the storm stayed at the ending or beginning stage of the lifecycle and its intensity was too low to be classified as a TC level. Second, there are no TCs involved in the image, but the cloud features are too similar to those for TC-images that the model failed to distinguish the two types correctly. Because the NWP image covers a much larger area than the L image, an NWP image tends to contain more such TC-like features than an L image,

which results in relatively more false positive predictions for the NWP images.

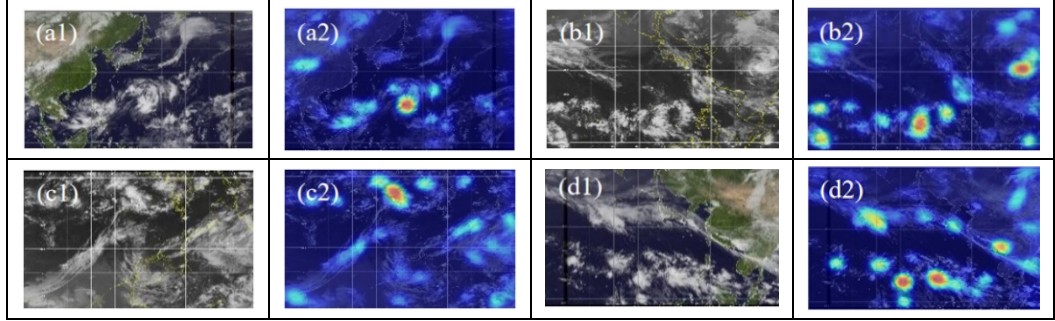

**Figure 16.** Non-TC images misjudged as TC images and associated heat maps

### 3.2.3 Dependence on TC intensity for SCIs with a single TC

To examine the dependence of prediction performance of the model on TC intensity, the testing samples of NWPO images involving a single TC were stratified into different groups according to the TC intensity. Figure 17 shows the

prediction probability for each of the intensity groups. Overall, the false positive predictions (or the recall ratio) become fewer (larger) as the increase of the TC intensity. Comparing the results to those shown in Figure 7, it is further found that there are fewer false negative predictions for NWPO images with TCs at low-to-moderate intensity levels (<70-80 knots), despite the fact that the number of testing samples for NWPO images is much larger than that for L images. The above findings suggest the proposed model herein has a good performance in terms of avoiding the omission of identifying TCs.

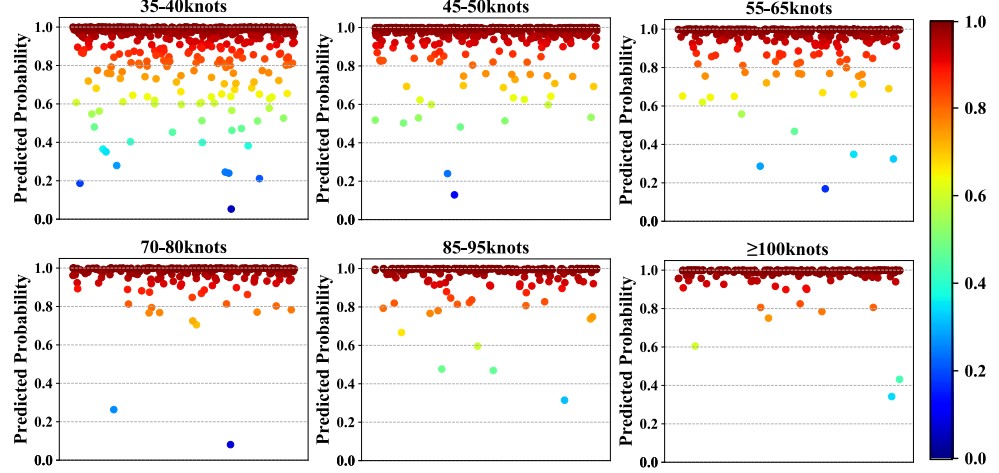


**Figure 17.** Prediction probability for NWPO images involving a single TC stratified by TC intensity





Figure 18 exhibits two types of successfully identified TC images and associated heat maps. As can be seen, the fingerprint features from both types can be reasonably focused on clouds. It seems that the model is able to distinguish

clouds from other background factors, such as coastal lines, continent/peninsulas, ocean surface and deserts which are depicted in varied colors in the image. The difference between the two types of results lies in that the main body of TC cloud in Type Ⅰ is identified as the primary fingerprint feature, while there are multiple fingerprint features highlighted in the heat maps from Type Ⅱ. The above difference may be explained by the varied characteristics of color gradation of the clouds between the two types of images. Results suggest that all fingerprint features with more concerns in the heat maps

correspond to brighter clouds in the image. In Type Ⅰ, the TC clouds are brightest among various kinds of clouds. By contrast, there are several cloud clusters in an image from Type Ⅱ whose color gradations are comparable to that of the TC cloud.

Figure 19 depicts some typical TC images that were misjudged as non-TC images. In reference to the images with a weak to moderately strong TC, the reasons to account for such false negative predictions are similar to those discussed for

Figure 8. For images with a strong TC (≥80 knots), the TC clouds are found to be distinctly smaller than their counterparts in Figure 18. It is expected that the IP structure of this paper fails to connect such tiny features with those in a normal scale. Thus, an IP structure with a bit more levels may be more appropriate for such cases.

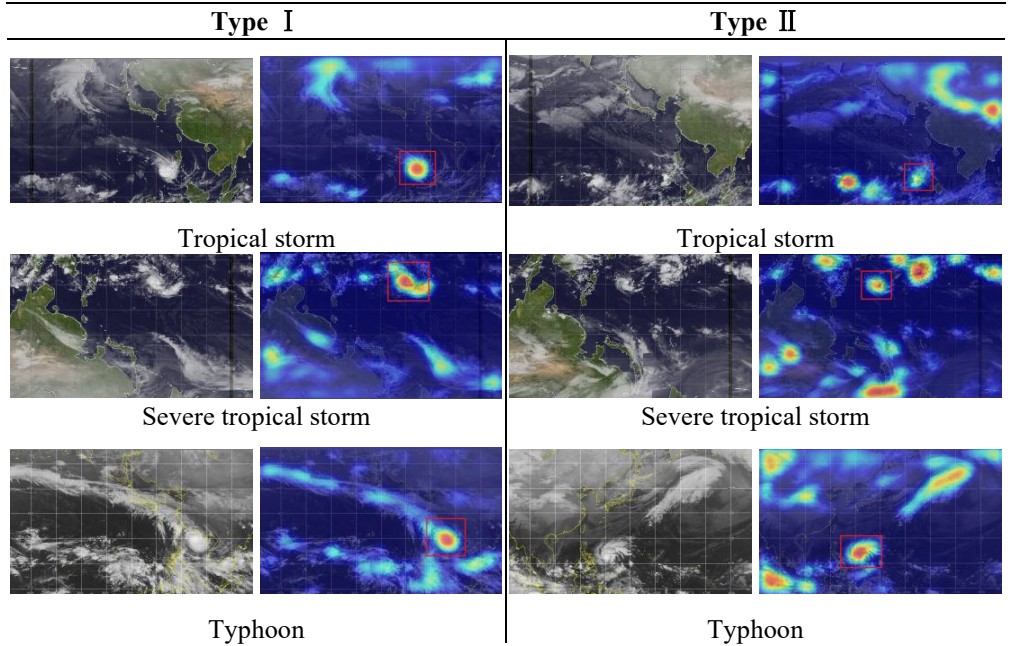



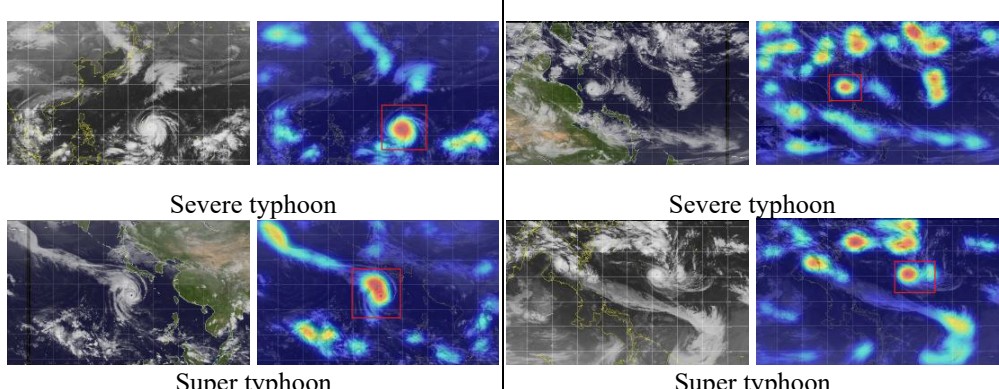

**Figure 18.** Heat maps of some successfully identified TC images compared with associated NWPO images that contain a single TC

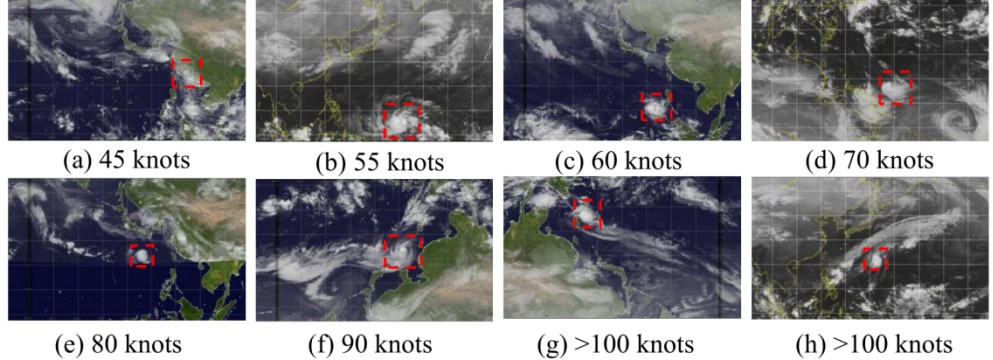

**Figure 19.** TC (demarcated by red rectangular) images misjudged as non-TC images

### 3.2.4 Dependence on the number of TCs within a TC image

One of the largest differences between L images and NWPO images is that there may be multiple TCs involved in an NWPO image, while there is no more than one TC in an L image. To examine the dependence of prediction performance of the model on the number of TCs within a TC image, the testing samples of NWPO images involving multiple TCs were stratified into different groups according to the TC number. Figure 20 shows the prediction probability for each of the groups. The recall ratios for the groups with an image involving one TC, two TC, and three or more TCs are 98.38%, 99.75 and 100%, respectively. It is clear that as the increase of TC numbers for associated NWPO images, it becomes more and more unlikely for the model to misjudge a TC image as a non-TC image. The above observation is consistent with one's expectation as an image involving more TCs usually contains more distinguishable fingerprint features that facilitate the model to predict more correctly.



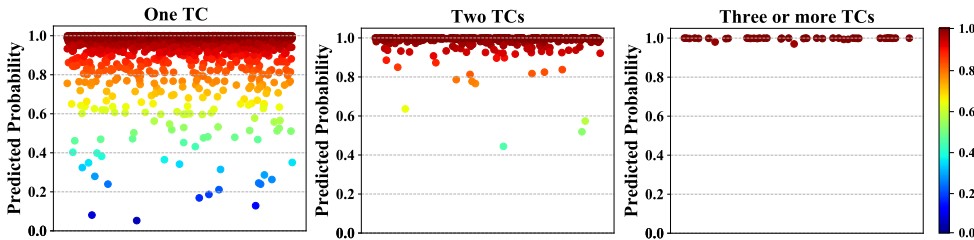

**Figure 20.** Prediction probability for NWPO images involving multiple TCs

Figure 21 examines the heap maps of some successfully identified images with multiple TCs. As demonstrated, all the
TCs involved in the images are highlighted in the heat maps from the two categorized types. For the results in Type Ⅰ, the
TCs are recognized exclusively as the primary fingerprint features. However, for the results in Type Ⅱ, there may be some
additional false targets identified; meanwhile, some of the TCs may receive even fewer concerns than the highlighted false
targets in the heat maps. From the perspective of TC identification, an NWPO image involving multi-TCs may be regarded
as a composite of multiple NWPO images, with each of them containing only a single TC. Consequently, the findings in
Figure 21 can be comprehended in a similar way to the one for Figure 18.

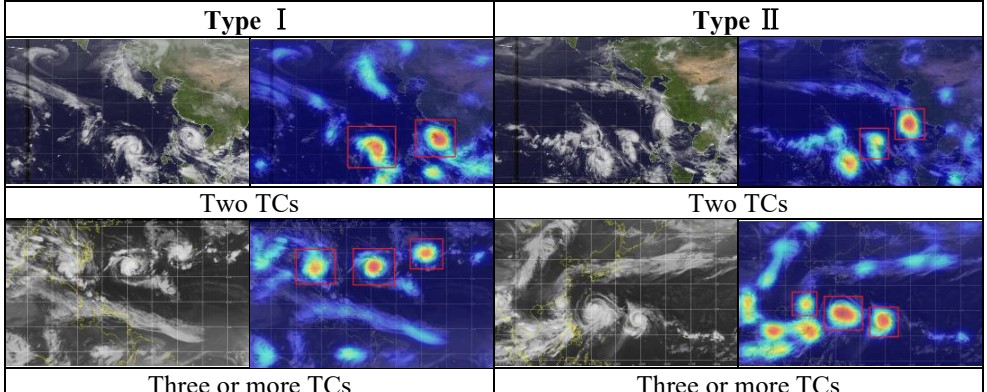

**Figure 21.** Heat Maps of some successfully identified TC images compared with associated NWPO images that involve
multiple TCs

## 5. Concluding remarks

This article presents a study on the identification of TC image from SCIs via DCNN techniques. Two DCNN models were
proposed to respectively deal with the issues associated with L images and NWPO images which covered different
geographical areas and varied numbers of TCs. Results suggested that the performances of the two models were satisfactory,
with both of the prediction accuracies exceeding 96%. Through analysis via heat map techniques, it was demonstrated that
the DCNN models are able to focus on TC-fingerprint features successfully during the testing process. Thus, it provides an
automatic and objective method to distinguish TC images from non-TC images by using deep learning techniques. This is



pretty useful for many SCI-based studies, e.g., SCI-aided identification of TC intensity, in which it is prerequisite to select TC images usually out from hundreds of thousands of SCI samples that correspond to both TC and non-TC conditions.

The two proposed models differ from each other by both the internal structures and the realization of training and validation processes. The NWPO model consisted of more convolutional layers and dropout layers so that it would be more efficient to extract useful information from a more complex image. More importantly, the IP techniques were specially adopted for the NWPO model to generate more appropriate training and validation datasets. Results show that the NWPO model failed to focus on correct targets if they were trained by conventionally pre-processed image samples in which the TC structures became considerably small with respect to the coverage area of the image. By contrast, when training the model by image samples pre-processed via the IP techniques, all TC fingerprint features could be identified correctly, which reflects the effectiveness of using the IP technique to deal with identification problems with small targets.

Despite the overall good performance, the DCNN models failed to provide correct predictions for some cases. Basically, there are three reasons to account for the failures. First, there are essentially no differences between an image involving a TC at a low intensity level and the one that should be regarded as a non-TC image but actually contains a tropical depression. Second, the morphological characteristics of TC clouds involved in some TC images, especially those corresponded to TCs staying at the very beginning and ending stages of their lifecycle, are too similar to those associated with non-TC conditions, which makes it to be challenge to identify such TC images correctly. Third, there were insufficient training samples for some special types of TC images (e.g., Figure 8(f)). Whatever the reason, more training samples that cover each type of the misjudged SCIs are needed so that the model can be trained to perform more accurately.

This study has only considered the identification problem, i.e., to judge whether a SCI belongs to a TC image or a non-TC image, but has not concerns with the problem where the TC is located in a TC image. Although all the main potential TC-fingerprint features have been identified in the heap maps, further efforts are needed to distinguish the true targets from the false.

*Author Contributions.* B. Tong performed drafting, methodology development and data analysis; X.F. Sun helped methodology development; J.Y. Fu operated conception, funding, editing and revision; P.W. Chan operated editing and revision; Y.C. He operated conception, drafting and revision, funding.

*Financial support.* This research was funded by the National Science Fund for Distinguished Young Scholars, China (No.51925802); the National Natural Science Foundation of China (No.51878194; No.52178465); the 111 Project (Grant No. D21021); the Guangzhou Municipal Science and Technology Project (Grant No. 20212200004).

*Acknowledgments.* The authors would like to thank our colleagues who made suggestions for our paper and the developers who selflessly provided the source code to the researchers. The data used in this study are openly available at the National Institute of Informatics (NII) at http://agora.ex.nii.ac.jp/digital-TC/, Meteorological Satellite Research Cooperation Institute,





University of Wisconsin-Madison (CIMSS) at http://tropic.ssec.wisc.edu/ and China Meteorological Administration tropical cyclone database (Ying et al., 2014; Lu et al., 2021) at http://tcdata.TC.org.cn/.

*Competing Interest.* The authors declare no conflict of interest.

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
