# Peer review of "Identification of tropical cyclones via deep convolutional neural network based on satellite cloud images"

_Atmospheric Measurement Techniques, 2021_

## Author Comment (AC1)

Joint Research Centre for Engineering Structure
Disaster Prevention and Control,
Guangzhou University
Guangzhou, China

13 January 2022

**In reference to amt-2021-405 "Identification of tropical cyclones via deep convolutional neural network based on satellite cloud images":**

The authors appreciate greatly the referee for his valuable comments and suggestions. We will address these concerns below.

**Comments from reviewers:**

-Referee #1

General comments:

**1. This paper focuses on the identification of TCs based on satellite cloud images via DCNN techniques. Two models are proposed to deal with identification issues associated with two kinds of SCIs that are widely utilized in this field. Visualization techniques are further adopted to examine how the DCNN models work internally. Overall, the article is well organized and written. Both the methodology (including datasets and models/methods) and main results are presented and discussed clearly and reasonably. The results are interesting and useful. This reviewer suggests the article be accepted after minor revision.**

**Response:** Thanks for the reviewer's comprehensive summary of this work and the encouraging comments. The manuscript has been revised carefully based on the received comments.

Specific Comments:

**2. Abstract: "lack of concerns on the identification of TC fingerprint from SCIs have become a potential issue, since it is a prerequisite step for follow-up analyses". Please revise this sentence to improve its readability, meanwhile, have may be replaced by has.**

**Response:** This has been replaced by "Although great achievements have been made in this field, there is a lack of concerns on the identification of TC fingerprint from SCIs which is usually involved as a prerequisite step for follow-up analyses" in the updated manuscript.

**3. L66: there lacks a blank**

**Response:** Revised accordingly.

**4. Line 131: tend--tends**

**Response:** Revised accordingly.

**5. Section 2.2.1: the authors use rotation technique for data augmentation. As discussed in this section, some information of the image may be lost. Will this operation result in any influence on the identification results?**

**Response:** Thanks for the meaningful comments. The authors agree with the reviewer's opinion that TC images generated via rotation manipulations will lose some information, but this operation should result in insignificant, if any, effects on the prediction performance of the proposed model. The reasons are given as follows.

Usually, a DCNN model consists of millions of coefficients which should be quantified reasonably during the training process. Thus, it is important for the model to have sufficient training samples to account for various types of issues. Unfortunately, there are usually insufficient samples in practices. To this end, rotation techniques are often adopted in the field of image identification. By adopting this technique in this study, two benefits can be achieved: (a) there will be more samples, which is greatly helpful for the identification performance of some image-types associated with limited samples; (b) the generalization ability of the model can be improved effectively.

Although TC images generated via rotation operations will lose some information, it does not mean that such operations will result in degraded performance of the model. After all, AI techniques may work in a quite different way with human beings, and many factors exist which can be adopted by the proposed model to provide acceptable prediction results. Actually, the overall performance of the DCNN models can be examined directly and objectively based on the identification results obtained during the testing stage. Results presented in the manuscript demonstrate that the DCNN model does performance well in terms of prediction accuracy. To further show that rotation techniques will not degrade the model performance, Table 1 compares the overall performance of the proposed model during the testing processes that are respectively based on TC images without rotation operations and those after rotation manipulations. It is seen that there is nearly no difference between the two kinds of results.

Table 1. NWPO image prediction performance of the TG-2 model during testing process

| Parameter | Accuracy | Precision | Recall | F1-Score |
|---|---|---|---|---|
| Result of the image rotation | 97.23% | 96.11% | 98.13% | 97.26% |
| Results of the image is not rotated | 97.82% | 97.96% | 97.80% | 97.92% |

**3. Line 148: have—has**

**Response:** Revised accordingly.

**4. Line 174: to judging**

**Response:** Revised accordingly.

**5. Lines 178, 182, 208: format (especially for where)**

**Response:** Thanks for pointing out this typo. Revised accordingly.

**6. The authors proposed two DCNN models. Although associated prediction results seem to be quite good, how about the comparative performance of these models against others?**

**Response:** Thanks for the useful comments. In fact, we have compared the performance between the model proposed in this study and other classification models (e.g., VGG16, ResNet50). The specific evolutionary curve and model comparison results are listed in Figure 1 and Table 2 (take L image for example). Results show that the stability of our model is slightly higher than the other two models, and the overall performance is also better than that for the other two models. Because this article focuses more on how to use the proposed model to identify TC images, we have not presented the comparison results.

[Figure]

Figure 1. Evolutional curves of the prediction accuracy of three DCNN models for L images

Table 2. L image prediction performance of the three model during testing process

| Parameter | Accuracy | Precision | Recall | F1-Score |
|-----------|----------|-----------|--------|----------|
| Our model | 93.38% | 90.12% | 98.22% | 94.00% |
| ResNet50 | 88.75% | 86.11% | 93.85% | 89.81% |
| VGG16 | 88.94% | 89.37% | 91.70% | 90.52% |

**7. The authors report two types of heat maps which vary with each other evidently. Are there any reasons for why there will be such two kinds of heat maps?**

**Response:** Several potential reasons are given as follows:

(i) There are indeed some patterns of features that can be only recognized by the DCNN model, and these features are quite different from those to which human beings are familiar.

(ii) DCNN models work in a quite different way from human beings. It seems that they only focus on whether the predictions are accurate, but do not concern if the prediction methodology is reasonable. It is possible that for some samples, DCNN models just make correct prediction results, but the methods (i.e., heat maps) are not

reasonable.

(iii) The working performance of the DCNN model depends greatly on the quality of SCIs and associated label information. As discussed in the article, some label information provided by meteorological institutes may not be accurate. The inaccuracy of such information results in abnormal features in associated heat maps.

(iv) It remains a challenging work to explore how network works internally, and current visualization techniques are not good enough to provide perfect heat map results.

**8. Section 3.2.1: it seems that to use the IP technology the authors have to extract zoom-in view of TCs from the NWPO picture If it is the case, how to do this?**

**Response:** Thanks for the meaningful comments. The image pyramid is random clipped according to the best TC tracking data. Firstly, the TC in the NPWO image was located using the best latitude and longitude provided by China Meteorological Administration. Then, we selected some TCs randomly, and extracted associated TC images according to different proportions. Measures were also adopted during the extraction process so that the proportions among pictures with large, medium and small scales are basically 1:1:1. In addition, non-TC medium and small-scale images are randomly cropped from large-scale samples with non-TCs, so that TCs would not appear in these images. Finally, we mixed these images with three scales together for training and validation.
* * *
The authors would like to express their sincere acknowledgement again for the reviewers' pertinent and insightful comments on this manuscript, which are much helpful for the improvement of the quality of it.

Sincerely yours,

Dr. Y.C. He (Associate Prof.)
Guangzhou University
E-mail: yuncheng@gzhu.edu.cn

---

## Author Comment (AC2)

Research Centre for Wind Engineering and
Engineering Vibration, Guangzhou University
Guangzhou, China

11 February 2022

**In reference to amt-2021-405 "Identification of tropical cyclones via deep convolutional neural network based on satellite cloud images":**

The authors appreciate the referee for his/her valuable comments and suggestions. We will address these concerns below by first quoting the comments.

**Comments from Referee #2**

General comments:

**1. In this study, deep convolutional neural network (DCNN) is adopted to identify TC satellite images. Efforts are also made to explore how the DCNN models work internally. Overall, the work is interesting, and the manuscript is well written, with analyses/discussions presented comprehensively and reasonably. I think this is a good piece of work which can contribute to existing literature. I have only some minor comments. It is suggested the article be accepted after minor revision.**

**Response:** Thanks for the reviewer's constructive suggestions as well as the encouraging comments. According to your comments, we have revised the manuscript. Detailed responses are stated as below.

Specific Comments:

**2. Line 24: ever--every**

**Response:** Revised accordingly.

**3. The authors claim that the normalization of image pixel values can accelerate the model convergence. Why?**

**Response:** To answer the above question more explicitly, we may take the following regression issue as an example. For equation $f_\theta(x) = \theta_1 x_1 + \theta_2 x_2 + b$, given two sets of values for variables $x_1$ and $x_2$ which are respectively in the range of [0,100] and [0,1], our aim is to determine the optimal values of coefficients $\theta_1$ and $\theta_2$. Let's assume that the influence of $x_1$-related item and $x_2$-related item in the equation on estimation of $\theta_1$ and $\theta_2$ is equal, which is usually the case in the field of image identification, then $\theta_2$ should be larger than $\theta_1$. By convention, the gradient descent method is adopted to estimate the optimal values of the two coefficients. To minimize the difference between training results and true values, one has to compute the derivative of $\theta_1$ and $\theta_2$. Since $x_1$ is greater than $x_2$, it can be easily deduced from the derivation formula that the descent speed of $\theta_1$ is much larger than that of $\theta_2$.

[Figure]

(a) without normalization (b) with normalization

Figure 1. Schematic diagram of optimization process via gradient descent method

Figure 1 shows a schematic diagram of the optimization process via the gradient decent method both without (a) and with (b) normalization technique. For Fig. 1(a), as the value ranges of $\theta_1$ and $\theta_2$ vary significantly, the gradient vector (marked as red arrow line) computed based on variable records at one step may not be in parallel with the one computed based on those at neighboring steps (i.e., demonstrating a zigzag pattern), which makes the optimization process to be comparatively longer and more time-consuming. By contrast, for Fig. 1(b), because all variable records are normalized to be in the range of [0 1], the values of $\theta_1$ and $\theta_2$ become in a similar range, so does their derivatives. As a result, the gradient vector computed based on variable records at one step tends to be in parallel with the one computed based on those at neighboring steps, which makes the optimization process to be shorter and more time-saving. Overall, the normalization technique is beneficial for speeding-up the model convergence.

**4. Please explain a little more about the Dropout layer.**

**Response:** The authors have added one more statement about how a Dropout layer works in the updated manuscript (lines 161-162): "During training, the dropout layer can randomly drop neural units from the neural network."

In a neural network, one layer is called the dropout layer because some of the neurons are removed from the neural network. To explain the above point more clearly, Figure 2 depicts a schematic diagram of the internal structure of a dropout layer. As can been seen, some neurons are disconnected with others, as a result, they seem to be dropped out from the network system. The Dropout layer is usually involved in the following steps during the training process:

  a) Set the dropout rate of each Dropout layer;
  b) Remove part of the neurons according to the corresponding rate before the training, and update online neurons / weight parameters during the training process;
  c) After all parameters are updated, some neurons are removed again according to the corresponding rate, and then the training begins;
  d) Repeat the above process until an acceptable fitting result is achieved.

In general, the larger and deeper the neural network is, the more likely it tends to suffer from over-fitting problems. In this regard, owing to the operations of dropout layer, some neurons can be randomly removed from the network, which is pretty useful for preventing over-fitting problem and for improving the universality of fitting results.

[Figure]

Figure 2. Schematic diagram of the structure of a Dropout layer

**5. Results of the evolutional curve in Figure 5(b) suggest that the training accuracy is not improved consistently with the increase of training epochs. When should you finish the training process?**

**Response:** Thanks for the meaningful comments. Based on our practical experience, it is not always the case that training accuracy is improved in trend with the increase of training epochs. Sometimes, training accuracy may decrease slightly and slowly with the increase of training epochs, as the one shown in Figure 5(b) in the manuscript. Generally, when the values of loss function become small enough (i.e., lower than a certain value) and change insignificantly with the increase of training epochs, the training process can be stopped. Our experience suggests that the best training results can be achieved within 80-100 epochs. Thus, in the presented manuscript, the model has been trained for 100 epochs. There are three points to be stressed.

(1) In practice, the training process would be stopped automatically according to the results of loss function. Although there are 100 epochs during the training process for the example show in Figure 5(b), the training process would have finished within the first 40-50 epochs in practice. The training process lasted for 100 epochs only because we forced it to do so.

(2) In this study, a method is adopted to retain the training information with the best validation accuracy, i.e., even that the training process lasted for 100 epochs in the presented example, only the parameterization information associated with the best training epoch has been retained, or more explicitly, the model after training for 100 epochs would be the same with the one trained for 50 epochs in this example.

(3) There are other embedded methods to determine when the training process would be stopped. For example, the training can be stopped in advance if the validation accuracy or loss is not improved within 10 epochs.

**6. The authors present many heat maps of TC images. How about those of non-TC image? Are there any typical differences between these two types of heat maps?**

**Response:** Thank you for your useful comments. Non-TC samples also have heat maps, which may differ from those of TC images significantly. An obvious difference between these two types of heat maps lies that there is a lack of massive concerning (or weighted) area(s) or the concerning areas are dispersedly distributed in the heat maps for non-TC samples, as demonstrated in Figure 3. In addition, for some heat maps corresponding to NWPO images with non TCs, there are relatively more concerns with onshore clouds, as shown in Figure 3 (c, d). However, sometimes, the two types of heat maps may demonstrate similar characteristics, as those discussed for Figure 11 and Figure 16 in the manuscript. As this article focuses on the identification of TC images, we have not discussed too much about the typical characteristics of heat maps associated with non-TC images.

[Figure]

(a)     (b)        (c)        (d)

Figure 3. Heat maps of non-TC images: (a) L image I; (b) L image II; (c) NWPO image I; (d) NWPO image II
* * *
The author sincerely thanks the reviewer for his kind advice and meaningful comments, which are valuable in improving the quality of our manuscript.

Sincerely yours,

Dr. Y.C. He (Associate Prof.)
Guangzhou University
E-mail: yuncheng@gzhu.edu.cn

---

## Author Comment (AC3)

Research Centre for Wind Engineering and
Engineering Vibration, Guangzhou University
Guangzhou, China

13 February 2022

**In reference to amt-2021-405 "Identification of tropical cyclones via deep convolutional neural network based on satellite cloud images":**

The authors appreciate the referee for his/her valuable comments and suggestions. We will address these concerns below by first quoting the comments.

**Comments from Referee #3**

General comments:

**1. This paper presents a research on classification of TC and non-TC pictures from satellite cloud images using the Deep convolutional neural network. Two image sets are used: the image set that covers all the Northwest Pacific Ocean basin with multi TCs, and L image set that covers small region of NWPO with single TC. The images are break out into training, validation and test sets.**

**Two DCNN models are trained for the two sets respectively. For the model trained with larger size images with multi TCs, the image pyramid technique is used to pre-process the images before training the model. Appropriate performance parameters are employed to evaluate the adequacy of the models. It shows that the pyramid technique improves the accuracy of the model.**

**The structures of the DCNN are well designed and presented in the paper. The results are well analyzed with proper discussion. The findings in this research should be valuable for further researches on this aspect, and even the models could be a useful basis for the meteorological agents to build their operational model on.**

**Response:** We thank the reviewer for the comprehensive summary of this work as well as the encouraging comments. The manuscript has been revised according to the received comments.

Specific Comments:

**2. Line 190ï¼ "Based previous tests" should be "Based on previous tests"**

**Response:** Thanks for the careful revision. Revised accordingly.

**3. Below section head 3.1.1ï¼ What does "The 10-fold cross-validation" mean? How is the "10-fold cross-validation" operated?**

**Response:** Cross validation has been introduced briefly in the manuscript (Lines 229-231), as follows:

"On the other hand, to improve the robustness of the model performance, the cross-validation strategy (Kohavi, 1995) is often exploited. As introduced previously, the original data in this study are stratified into 10 parts, with 9 parts used as

training/validation set and 1 part as test set. By using the cross-validation strategy, the data can be trained and tested upmost for 10 times".

Actually, cross validation is used as a standard technique in the field of machine learning. Thus, besides the above brief introduction, only a related reference is provided and cited in the text. We may detail how a cross-validation process is conducted through the following example.

Let's take 10-fold cross-validation for the L images as an example. After pre-processing, there are totally about 32,000 samples. We can randomly divide these images into 10 parts, with each part containing about 3,200 images. We then label the 10 parts with serial numbers from 1 to 10. After that, we can first take Parts 1-8 as the training set, Part 9 as the validation set, and Part 10 as the test set. As discussed in the manuscript, both training set and validation set are used during the training process, while the test set is utilized for examining the performance of the model. After the above operation, we can take Parts 2-9 as the training set, Part 10 as the validation set, and Part 1 as the test set, and to train and test the model. Similar operations can be repeated 10 times. We can then examine the robustness of the model performance based on 10 times of testing results.

From the above example, it can be seen that the 80% of training data for each of the 10 operations are selected randomly, and the distribution of the training/test set was also random. To sum, cross-validation technique is adopted to guarantee that the model is able to work robustly.

**4. In Figure 5, what is the difference for TG-1 to TG-10? Any difference in parameter setting among them? The same question for TG-1 to TG-3 in Figure 14.**

**Response:** The training, validation and test datasets involved in models TG1-to-TG10 are different, but other hyperparameters are the same. As stated in the response to Comment-3 about the cross-validation technique, in order to comprehensively evaluate the robustness of the model and to avoid potential influence caused by the contingency of data distribution, we train and validate the network for 10 times, leading to 10 parameterized models. It is clear that these models tend to be different in specific values of involved parameters/coefficients, so are the learning curves. In addition, random descent method is adopted to finalize the optimization function, which may also result in certain differences in the internal parameters of each model. Once such differences are significant, they will be reflected in the learning curve (such as over-fitting and under-fitting).
* * *
The author sincerely thanks the reviewer for his kind advice and meaningful comments, which are valuable in improving the quality of our manuscript.
Sincerely yours,

Dr. Y.C. He (Associate Prof.)
Guangzhou University
E-mail: yuncheng@gzhu.edu.cn